# The Effect of Polydopamine on an Ag-Coated Polypropylene Nonwoven Fabric

**DOI:** 10.3390/polym11040627

**Published:** 2019-04-04

**Authors:** Chuanmei Liu, Jie Liu, Xin Ning, Shaojuan Chen, Zhengqin Liu, Shouxiang Jiang, Dagang Miao

**Affiliations:** 1Industrial Research Institute of Nonwovens and Technical Textiles, College of Textiles and Clothing, Qingdao University, Qingdao 266000, China; chineselcm@163.com (C.L.); jliu0411@163.com (J.L.); xning@qdu.edu.cn (X.N.); qdchshj@126.com (S.C.); zhengqinliu@126.com (Z.L.); 2Institute of Textiles and Clothing, The Hong Kong Polytechnic University, Hong Kong, China; kinor.j@polyu.edu.hk

**Keywords:** magnetron sputtering, dopamine, stability and durability, multifunction

## Abstract

A practical method for preparing multifunctional polypropylene (PP) nonwoven fabrics with excellent stability and durability was explored. First, the PP nonwoven fabric was sputtered by a magnetron sputtering system to form an Ag film on the surface of the fabric. Subsequently, the coated fabric was treated with dopamine. The fabrics were characterized by scanning electron microscopy (SEM), an energy dispersive spectrometer (EDS), electrical conductivity, electromagnetic interference shielding effectiveness (EMI SE), antibacterial activity, stability, and laundering durability. The results of the study revealed that the fabric was coated with Ag, and after the treatment with dopamine, the surfaces of Ag-coated fibers were coated with polydopamine (PDA). The fabrics still had a sheet resistance below ~15 Ω/sq and exhibited excellent EMI SE above ~25 dB, though few differences existed from the single Ag-coated sample. After the treatment with dopamine, the antibacterial activity of the fabric was enhanced. Meanwhile, the treated samples exhibited excellent resistance against sodium sulfide corrosion, which could enhance the stability of the Ag-coated fabric. Moreover, the laundering durability of the treated fabric was improved in the same process, whose lowest sheet resistance was ~18 Ω/sq and the EMI SE was ~8 dB more than single Ag-coated PP nonwoven fabrics. In conclusion, this method was considered to be effective in fabricating multifunctional, stable, and durable fabrics.

## 1. Introduction

In recent years, an increasing number of research studies have focused on surface modification, which is closely connected to the design and application of advanced materials [1,2]. Additionally, research scholars are dedicated to combining nanotechnology with synthetic fibers to produce multifunctional materials [3]. Typically, nanoparticles deposit on a fabric as a coating that endows the fabric with unique properties, such as UV protection, water repellency, bacteriostatic, and high conductivity [4,5]. Atomic layer deposition, layer by layer assembly, electroless deposition (ELD) [6], nanomaterial in situ deposition, and magnetron sputtering technology have been used to fabricate multifunctional fabrics [7,8,9,10]. Among these methods, magnetron sputtering technology has attracted significant attention from the scientific community because of its novel functionalities, such as its environmental friendliness, low processing temperature, and high deposition speed [11,12]. However, the low adhesion, stability, and durability of the coating have become the vital obstacles of magnetron sputtering technology, thus becoming the application limitation of this technology [13,14].

This problem has been solved by pretreatment, such as surface etching, which is limited by the type of substrates, and finishing treatment. Chemical finishing techniques are extensively employed to change the specific functional properties and improve their aesthetic values, which impart desirable properties, such as anti-wrinkle, flame-retardant, antistatic, and antimicrobial characteristics [15]. Jiang et al. [14] introduced a finishing process to improve adhesion and wear durability at the interface of a copper (Cu) film and a polyester fabric. However, stability and laundering durability were not explored.

Marine mussels have a superior capability of tightly attaching to rocks and ships in seawater through secreted adhesive proteins [16,17,18]. Inspired by this extraordinary adhesive behavior, dopamine was reported to form thin, surface-adherent polymer films on virtually all material surfaces, in alkaline aqueous media. Messersmith et al. [19] reported that PDA coatings can be facilely constructed on various substrates by the oxidized self-polymerization of dopamine in alkaline solutions, which exhibits numerous prominent properties including easy-to-implement and strong binding forces to almost all kinds of substrates’ surfaces.

According to Lee et al. [16], the polymerization of dopamine results in a sticky layer that can uniformly coat a wide variety of substrates, including oxides, polymers, and ceramics. The PDA coating layer is fairly stable, even under ultrasonic treatment [20] or over a long exposure time [21]. Up to now, this mussel-inspired coating strategy has been regarded as a promising approach to surface modification in many fields, such as those of bioengineering [22,23,24], energy [25], and environmental science [26,27,28]. As for the metal nanoparticles, scientists usually focus on the reducing ability of PDA toward metal ions, forming metal nanoparticles [29]. They seldom use the adhesive property of PDA [30,31].

Nowadays, a fabric with an Ag nanoparticle coating using magnetron sputtering technology has drawn attention due to its unique properties, such as its very low resistivity among all metals and antibacterial activity. Thus, an Ag film is the optimal choice for fabricating multifunctional materials. Nevertheless, Ag nanoparticles can be easily oxidized in air, which has been a non-negligible obstacle to conductivity. Moreover, the application of Ag was restricted due to the poor adhesion between the Ag coating and substrate [32,33,34], especially when washed in water. After washing, the properties of the coated samples were reduced to a large extent, or even disappeared.

However, there is insufficient literature on the post-treatment of Ag-coated fabrics for the improvement of the film’s stability and durability. This project, the extraordinary adhesive of PDA, was firstly used to improve the stability and durability, especially the laundering durability, of the Ag coating, which is fabricated by magnetron sputtering technology.

In this paper, a PP nonwoven fabric is chosen as the substrate because of its special continuous surface structure that can be conductive after sputtered, and stable chemical properties, especially the alkali resistance. First, the PP nonwoven fabric was coated with an Ag film by magnetron sputtering. Subsequently, the fabric was immersed in a dopamine solution, to form the PDA coating on the Ag-coated fabric. To fully investigate the prepared samples, analysis was performed using scanning electron microscopy (SEM), an energy dispersive spectrometer (EDS), electrical conductivity, electromagnetic interference shielding effectiveness (EMI SE), antibacterial activity, stability, and laundering durability.

## 2. Experimental

### 2.1. Materials

Chemicals: 3,4-Dihydroxyphenethylamine (dopamine) was purchased from Aladdin (Shanghai, China) (98%)and used as received. Tris (hydroxymethyl) aminomethane (Tris) was purchased from Aladdin (Shanghai, China) (99%). Sodium sulfide was purchased from Xilong Technology Co., Ltd. (Shantou, China) (98%). All the chemicals used in this study were of analytical grade and used without any further purification process.

Substrates: Polypropylene nonwoven fabrics (36 g/m^2^, SMS) were purchased from CHTC JIAHUA NONWOVEN Co., Ltd. (Xiantao, China). Before being used, PP nonwoven fabrics were scoured with acetone for 30 min to clean them and then fully washed by deionized water. Later, the as-prepared fabrics were dried under 60 °C for 30 min.

Targets: Ag (99.99%) was purchased from Zhongnuo New Materials Technology Co., Ltd (Beijing, China).

### 2.2. Magnetron Sputtering

The Ag film was sputtered on one side of the PP nonwoven fabrics at room temperature by magnetron sputtering apparatus. The Ag film was prepared by DC magnetron sputtering of the Ag target. To improve the uniformity of the coated film, the sample holder was rotated at a speed of 10 rpm. The deposition parameters are presented in Table 1. The thickness of the external surface coating was measured and controlled by an online FTM-V film thickness monitor (TAIYAO Technology Co., Ltd., Shanghai, China).

### 2.3. Deposition of PDA on PP Nonwoven Fabrics

The Ag-coated PP nonwoven fabric was first prewetted by ethanol and then immersed in the Tris buffer solution (pH = 8.5, 50 mM) for several hours at a static state. Subsequently, dopamine hydrochloride was dissolved in the Tris buffer to make a dopamine solution (2 mg/mL), and the samples were infiltrated in the solution for 3 h, 6 h, 9 h, and 12 h at 50 °C. Then, the samples were washed with deionized water several times to move any labile PDA, if any, on the surfaces of the fabrics and dried in vacuum at 60 °C for 0.5 h. The schematic of the deposition process of the PDA coating is illustrated in Figure 1.

Figure 2 illustrates the structure of dopamine and PDA. In the alkaline and oxygen condition, the catechol group in dopamine is oxidized to dopamine quinone. The structure of dopaminequinone is erratic and could be further oxidized into many intermediates, such as leukodopaminechrome, dopaminechrome, and 5,6-dihydroxyindoleand. Intermediates then experienced intermolecular and intramolecular rearrangement, and cross-linking to form the PDA coating [35].

### 2.4. Characterization

Surface morphology of the treated fabrics was observed using the Phenom Pro Scanning Electron Microscope (SEM) (JEOL Ltd., Tokyo, Japan) operated at an acceleration voltage of 10.0 kV. For all the samples, the experiments were repeated three times, and the SEM images presented in the manuscript represent all repeated experiments.

The EDS analyses were carried out using an JSM-7800F Scanning Electron Microscope (JEOL Ltd., Tokyo, Japan) coupled to a X-Max Energy-Dispersive X-ray spectrometer (JEOL Ltd., Tokyo, Japan) for electron image acquisitions and elemental analysis (punctual and imaging), respectively.

The electrical conductivity of the treated samples was measured using a four-point probe system ST-2258C (Suzhou Jingge, Suzhou, China). Every resistivity value of the sample was the mean of the five measurements that were made in different places of the same sample.

The EMI SE of the treated fabrics was measured in the frequency range of 0–3 GHz using a fabric anti-electromagnetic radiation performance tester DR-913G (DARONG Textile Instruments Co., Ltd., Wenzhou, China) at room temperature.

The antibacterial activity of the treated fabric was qualitatively assessed against gram-positive bacteria *Staphylococcus aureus* (*S. aureus*) and gram-negative bacteria *Escherichia coli* (*E. coli*), according to the AATCC 90 method. All plates were examined for the inhibition zones after 24 h of incubation.

### 2.5. Stability

The stability of the treated fabric was evaluated by the treatment with sodium sulfide. The treated fabrics were completely immersed in the sodium sulfide solution (20 mg/L) for 30 min and then washed with deionized water. After that, the fabrics were dried at 60 °C for 30 min. The electrical conductivity and EMI SE were then measured.

### 2.6. Durability

The durability of the treated fabrics was evaluated by washing the fabrics using a method specified by the American Association of Textile Chemists and Colorists (AATCC) Test Method 61-2006 Test No. 1A. During washing, the samples were washed using a standard Launder-ometer (SW-20B, Quanzhou Meibang Instrument Co., Ltd., Quanzhou, China) equipped with stainless-steel lever-lock canisters (75 × 125 mm). The treated fabrics (50 × 100 mm) were immersed in a 200 mL aqueous solution containing a 0.37% (*w*/*w*) AATCC standard reference detergent without an optical brightener and 10 stainless steel balls. During laundering, the temperature was controlled at 42 °C, and the stirring speed was 40 ± 2 rpm. After 45 min of laundering, the laundered fabrics were washed with deionized water, and then dried at 60 °C for 30 min. The electrical conductivity and EMI SE were then measured.

## 3. Results and Discussion

### 3.1. Morphology and Chemical Composition Analysis

As we can see, the polypropylene fibers were intertwined irregularly, so that there were enough nodes between fibers, which could ensure the electrical conductivity of the fabric after the magnetron sputtering process. After the process of magnetron sputtering, the surface of the fibers was coated with a thin Ag film, and the Ag particles were deposited randomly on the surface, which we can observe from Figure 3a. The EDS mapping dots analysis of the C, O, and Ag elements obtained from the protrusion pattern fiber surface are shown in Figure 3b.

As shown in Figure 3b, the mapping dots of Ag elements tend to have a homogeneous distribution throughout the fiber’s surface where they were sputtered. The table in Figure 3b shows the proportion of the three elements, where the proportion of Ag is 61%, which proves that the fiber was coated with Ag. Furthmore, the proportion of C is 36%, which was obtained from the polypropylene fibers. Additionally, the oxidation of Ag resulted in the proportion of O of 3%.

Surface morphologies of the fibers with and without PDA coating are illustrated in Figure 4. The blank PP nonwoven fibers had smooth surfaces, as illustrated in Figure 4a. Based on the comparison of Figure 4a,b, it was found that there were no major changes in the morphology of the two surfaces at small magnification (×2000), except for the removal of the impurities that made the surfaces smoother. Figure 4c–f exhibit the morphology of the fibers with different dopamine treatment times. As illustrated in the SEM images, the surface of the fibers is prone to be rougher because of the deposition of the PDA particles. Moreover, the particle size ranged from 0 μm up to 0.490 μm as the treatment time increased, which is shown in Figure 5a. It can also be seen that the PDA coating is finally formed.

It is difficult to measure the increasing amount of PDA from only the changes in the morphology, but it can be quantitatively described through Figure 5a. The weight gain was presented by an equation:
$$M=\frac{M_{i}-M_{i0}}{S_{i}}$$
where M (g/m^2^) denotes the weight gain of the fabric; $M_{i}$ (g) is the the weight of the sample after the treatment of dopamine; $M_{i0}$ (g) is the weight of the sample before the treatmenet of dopamine; $S_{i}$ (m^2^) is the area of the sample; and i (h) is the treatment time 0, 3, 6, 9, and 12 h, respectively.

The value of the weight gain increased with the increasing treatment time of dopamine. However, it should be noted that the rates are not all the same: for the treatment time of 12 h, the weight gain was faster than those in other treatment times, which is consistent with the morphology analysis in Figure 4. The color change of the dopamine solution shown in Figure 5b confirmed that the color changed from colorless to pale brown and finally turned to deep brown with the process of oxidation.

### 3.2. EDS Analysis

EDS mapping dots analysis and the SEM image of the dopamine treated Ag-coated fabric are displayed in Figure 6. The mapping dots of C, O, N, and Ag elements are prone to having a homogeneous distribution. Compared with Figure 3b, the kind of elements, as well as the proportion, changed. The PDA possesses C, O, and N elements, so the element of N was totally abtained from PDA and it was distributed homogeneously on the surface of the fiber, which proved that the PDA distributed evenly on the fiber. Moreover, the amount of O was up to 18%, which was due to the O element in PDA and the Ag oxidation. Therefore, it can be easily concluded that the fiber was deposited with PDA. As shown in the SEM images, the PDA paritcles that were deposited onto the Ag film formed a PDA coating, and the particle size is about 0.49 μm for the treatment time of 12 h.

### 3.3. Electrical Conductivity Analysis

Ag possesses electrical conductive properties of approximately 1.59 × 10^−8^ Ω m at 293 K, whereas the PDA coating and particles can be regarded as an electro-insulator, and the deposition of PDA may affect the conductivity of the sample. As presented in Table 2, the dopamine-treated samples still had excellent conductive properties of about 10 Ω/sq, though higher than the sheet resistance of the single Ag-coated sample. It is obvious that the sheet resistance increased with the increasing treatment time of dopamine because of the obstruction of electronic transmission of the PDA coating. As illustrated in Figure 7, the LED connected with a piece of treated fabric in a circuit was lit when a voltage of 9.0 V was applied, thus indicating the satisfactory conductivity of the resultant fabric.

### 3.4. Electromagnetic Interference Shielding Effectiveness Analysis

Owing to its excellent electrical conductivity, the treated fabric can effectively reflect and absorb electromagnetic radiation, thus resulting in excellent EMI shielding performances.

The EMI SE of the untreated and treated samples is illustrated in Figure 8. As illustrated in Figure 8a, the EMI SEs of the blank PP nonwoven fabric and the PDA-coated PP nonwoven fabric are ~0 dB, which indicates a weak EMI SE of the two samples. The Ag-coated sample, whose EMI SE is ~26 dB, has excellent electromagnetic interference shielding properties.

The PDA-deposited Ag-coated samples had a much higher EMI SE than the blank PP nonwoven fabric. The EMI SEs are over 25.6, 25.1, 25.8, and 25.2 dB when the treatment time is 3, 6, 9, and 12 h, respectively. As illustrated in Figure 8b, the EMI SE property was a little lower than that of Ag-coated samples; this may have been caused by the influence of water immersion and blending when samples were immersed in the dopamine solution. The Ag film on the surface of the fabric was damaged to some degree. Through the EMI SE analysis, we can conclude that after the treatment of dopamine, the samples still had satisfactory EMI SE properties.

### 3.5. Antibacterial Activity Analysis

The antibacterial activities of the as-treated fabrics are elucidated in Figure 9. The zero inhibition zone and the microbial population on the control were observed [36]. There exists an obvious variation between the blank and Ag-coated fabrics. It is worth mentioning that the blank PP nonwoven fabrics do not have any antibacterial activity against *S. aureus* and *E. coli*. No bacteria colonies were found close to the edges of the Ag-coated fabrics, which was shown in Figure 9a, revealing that the Ag deposited on fabrics inhibited the growth of bacteria. Moreover, the widths of the inhibition zone (shown in Figure 9b) of the dopamine-treated (6 and 12 h) Ag-coated fabrics were measured to be 3.2 and 3.5 mm for *S. aureus*, and 3.9 and 5.5 mm for *E. coli*, respectively, which were larger than that of the single Ag-coated fabrics (2.1 and 3.0 mm in width for *S. aureus and E. coli,* respectively). Thus, this reveals that the treatment of dopamine can enhance the antibacterial activity of Ag-coated fabrics.

Ag nanoparticles can be easily oxidized into AgO, and the PDA has a strong reducing ability, so the AgO particles are reduced to Ag nanoparticles by PDA, which enhance the antibacterial activity of the fabric. It was apparent that compared with the inhibition zone of *S. aureus*, *E. coli* exhibited a higher inhibition zone. These differences are due to the differences in the cell-wall structure of the two types of bacteria.

### 3.6. Stability Analysis

Ag has stable physical and chemical properties, which contributed to it being widely used. However, Ag is highly susceptible to corrosion in sulfur and fluids [37]. In the present study, samples were treated in an Na_2_S solution. Then, the Ag-coated sample exhibited a great change in surface color, whereas the dopamine-treated Ag-coated samples changed slightly or did not change at all, which we can see from Figure 10. The SEM images of the samples after the treatment of Na_2_S are displayed in Figure 11.

As illustrated in Figure 11a, the single Ag-coated sample was seriously damaged. The Ag film attached to the surface was broken and peeled off piece by piece until the explosion of the sample’s original surface. Figure 11b–e denotes the samples with PDA deposition of different treatment times, whose Ag films remained in integral states, although they were slightly damaged. When the treatment time was up to 12 h, none of the differences could be observed in the sample in Figure 10e before the treatment with Na_2_S (Figure 4b).

The conductivity of the samples before and after the corrosion of Na_2_S is illustrated in Figure 12a. This image indicates that the electrical conductivity of the single Ag-coated sample declined rapidly and the sheet resistance was about 70 Ω/sq because of the broken Ag film. It is obvious that the sheet resistances of the samples after corrosion of the Na_2_S were closer to those of the samples before the corrosion with the increasing treatment time of dopamine. Additionally, when the time is up to 12 h, the sample has the lowest sheet resistance difference (approximately 2 Ω/sq) when compared with those at other treatment times of dopamine.

Figure 12b–f exhibits the EMI SE of the samples before and after the corrosion by Na_2_S with different treatment times of dopamine. The EMI SE of the single Ag-coated sample after corrosion was ~10 dB, which is illustrated in Figure 12b. It is worth mentioning that because of the existence of PDA, the corrosive effect of Na_2_S declined, which can be speculated from Figure 12c–f. The EMI SE of the samples is closer to these before corrosion with the increasing treatment time of dopamine and is almost coincident with that of the sample which was treated with dopamine for 12 h.

It is apparent that PDA can prevent the Ag layer from being seriously corroded by Na_2_S. The Ag coating formed metal coordination with the dopamine [38] and with the increasing of the treatment time, the surface of the Ag coating was covered with PDA coating, which can weaken the corrosion of Ag film. Therefore, the samples become much more stable because of the treatment of dopamine.

### 3.7. Laundering Durability Analysis

To evaluate the laundering durability of the treated samples, SEM images of the samples after washing are provided in Figure 13. It is indicated in Figure 13a,b that the single Ag-coated sample and the 3 h dopamine-treated sample were broken and even fell off the fibers, thus verifying the low adhesion of Ag films. With the increasing treatment time of dopamine, the Ag films preserved a high degree of integrity, as illustrated in Figure 13c–e. Therefore, the treatment of dopamine can improve the laundering durability of the Ag film from the analysis of surface morphology.

Figure 14 illustrates the electrical conductivity and EMI SE of the samples after washing. In Figure 14a, the sheet resistance of the Ag-coated fabrics which were treated with dopamine for 0 and 3 h could not be detected after washing. We can conclude that the Ag films have poor washing fastness because of their blending and folding during the washing process. After washing, when the dopamine treatment time is more than 3 h, the samples possess electrical conductivity and maintain better conductivity with the increasing treatment time of dopamine. When the treatment time is up to 12 h, the sample possesses the lowest sheet resistance of about 18 Ω/sq.

Figure 14b illustrates the EMI SE of the samples after washing. The EMI SE of the fabrics with dopamine treated for 0 and 3 h had poor EMI SE, which was almost down to 0 dB after washing. However, when the treatment time was more than 3 h, the EMI SE had not decreased sharply after washing. In addition, with the increasing dopamine treatment time, the samples had electromagnetic shielding performances which were all above 7 dB.

Consequently, considering Figure 13 and Figure 14, it can be concluded that the single Ag-coated sample has poor laundering durability. The Ag film would be badly damaged after washing. However, after the treatment of dopamine, the laundering durability can be improved compared with the single Ag coated sample, because the PDA can form a covalent bond with the fabric [18] and at the same time, it can also form coordination bonding and chelating bonding with the Ag coating, which means that the Ag coating is tightly adhered to the PP nonwoven fabric. Additionally, the excellent adhesion plays a vital role in improving the durability of the treated fabric. Also, the external PDA coating can protect the Ag film on the fabric from serious damage.

## 4. Conclusions

An innovative method was presented for the successful fabrication of a multifunctional fabric with an excellent stability and durability through magnetron sputtering and finishing treatment with dopamine. It was concluded that the fabric was coated with Ag, and with the increasing treatment time of dopamine, more and more PDA particles were deposited on the Ag-coated fabrics. The EDS analysis suggests the existence of PDA. After the treatment with dopamine, the fabric still exhibited excellent electrical conductivity of ~10 Ω/sq and EMI SE of ~25 dB. Moreover, because of the existence of PDA, the antibacterial activity of the fabric was enhanced. The treated fabric had a good stability when immersed in an Na_2_S solution. Moreover, the laundering durability of the treated fabrics was improved compared with single Ag coated PP nonwoven fabrics; when the treatment time was up to 12 h, the sample possessed the lowest sheet resistance of about 18 Ω/sq; and the EMI SE values were all over 7 dB. The treated fabric can be used as a smart fabric.

## Figures and Tables

**Figure 1 polymers-11-00627-f001:**
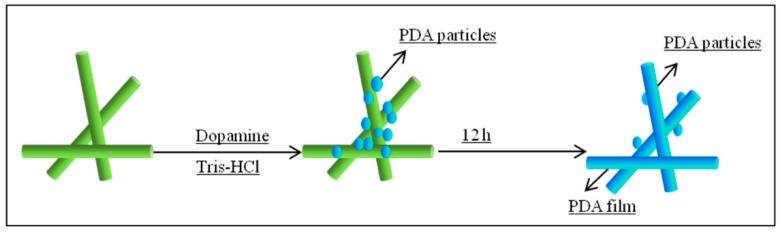
Schematic of the deposition process of PDA coating.

**Figure 2 polymers-11-00627-f002:**
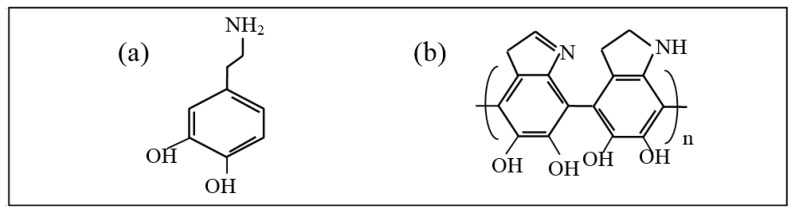
The structure of: (**a**) dopamine; (**b**) PDA.

**Figure 3 polymers-11-00627-f003:**
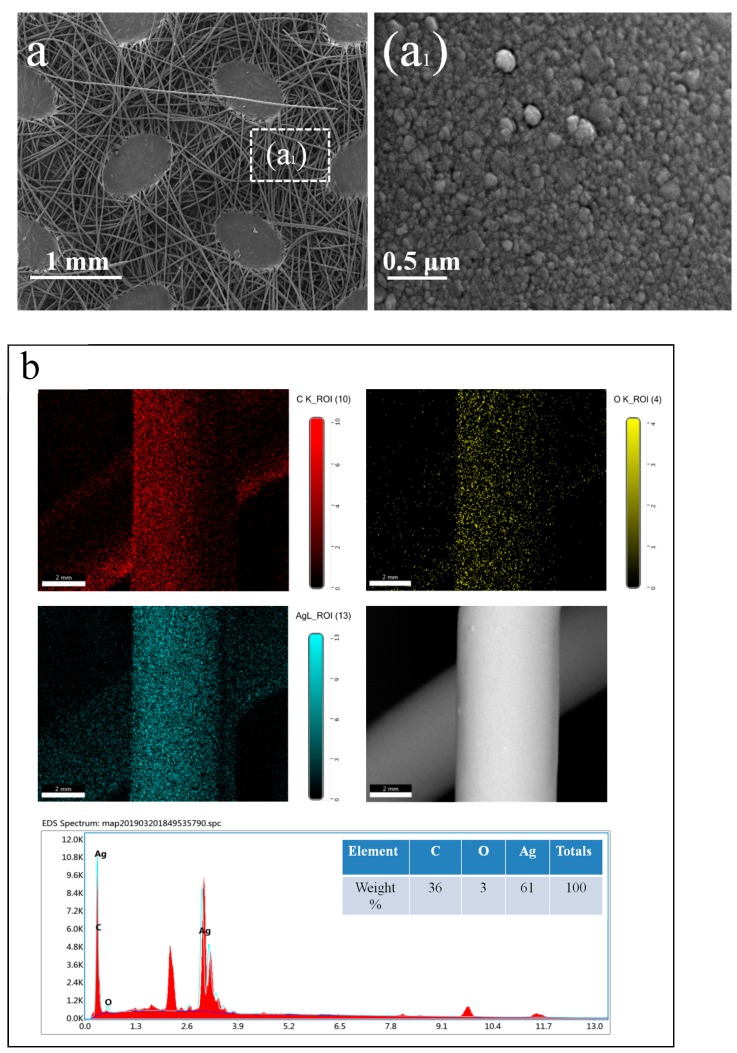
Surface morphology and chemical composition of Ag sputtered nonwoven fabric. (**a**) SEM images (50× & 40,000×) of Ag sputtered PP nonwoven fabric; (**b**) the elemental mapping image of the sputtered fabric.

**Figure 4 polymers-11-00627-f004:**
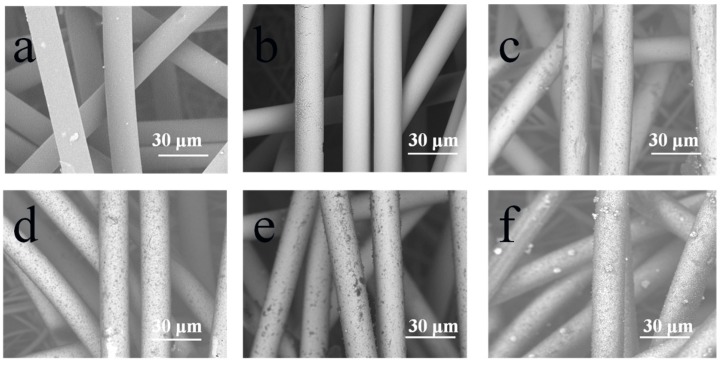
Surface morphology of fibers with PDA coating. (**a**) A blank PP nonwoven; (**b**) Ag-coated PP nonwoven; (**c**) PDA-treated (3 h) Ag-coated PP nonwoven; (**d**) PDA-treated (6 h) Ag-coated PP nonwoven; (**e**) PDA-treated (9 h) Ag-coated PP nonwoven; and (**f**) PDA-treated (12 h) Ag-coated PP nonwoven.

**Figure 5 polymers-11-00627-f005:**
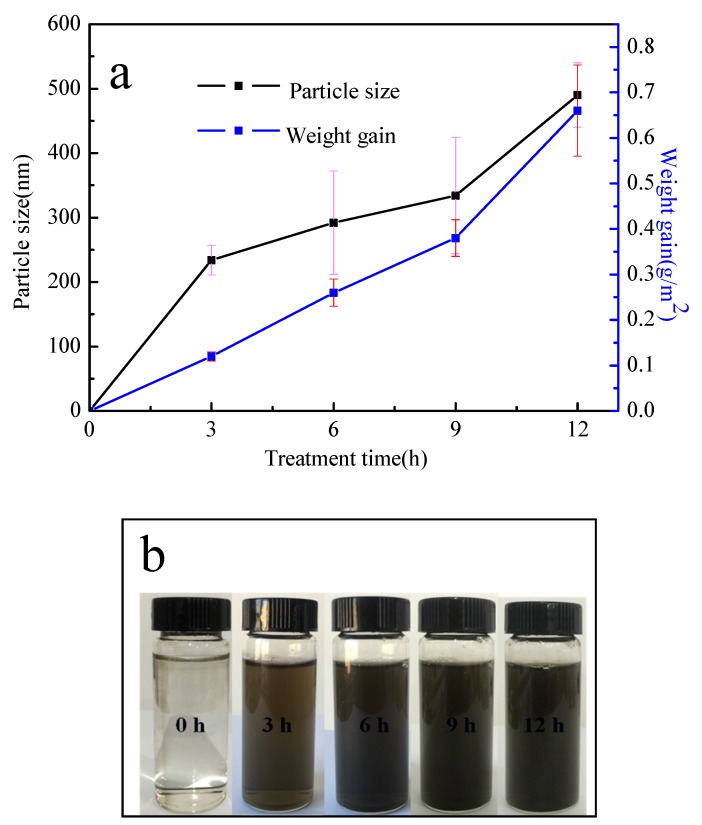
(**a**) The weight gain and the PDA particle size of the fabric with the treatment time of dopamine; (**b**) the color change of the dopamine solution.

**Figure 6 polymers-11-00627-f006:**
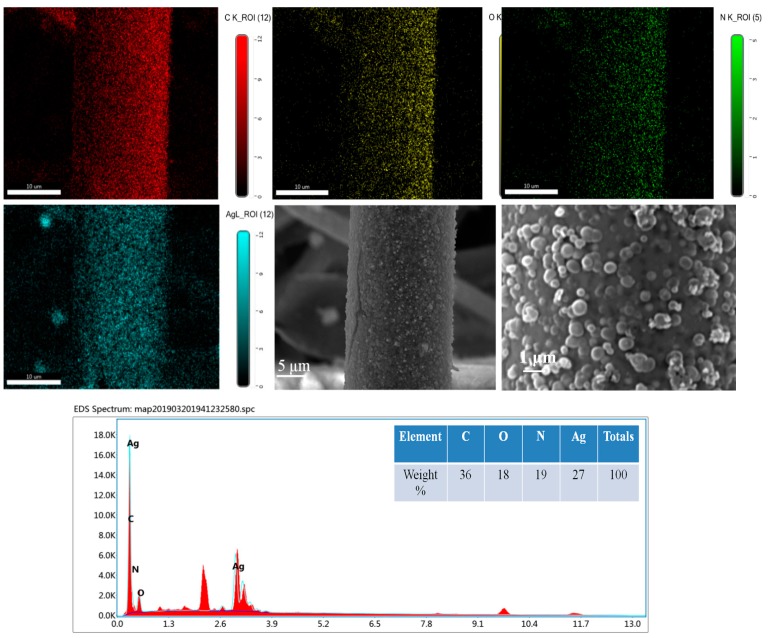
The mapping dots and elements’ weights of the dopamine-treated Ag-coated fabric.

**Figure 7 polymers-11-00627-f007:**
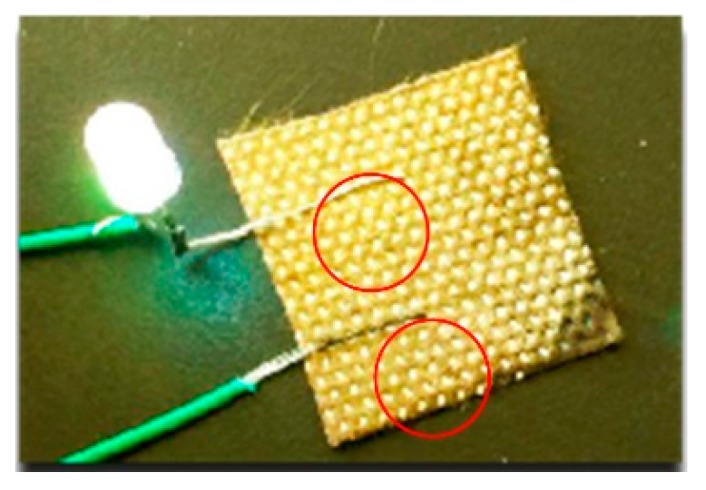
Photograph of the treated fabric (PDA 12 h) as a wire for powering an LED bulb.

**Figure 8 polymers-11-00627-f008:**
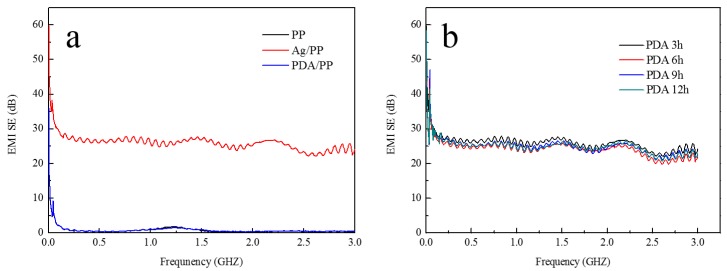
EMI SE of the treated and untreated samples: (**a**) samples before the treatment of dopamine; (**b**) samples after the different treatment time (3, 6, 9, 12 h) of dopamine.

**Figure 9 polymers-11-00627-f009:**
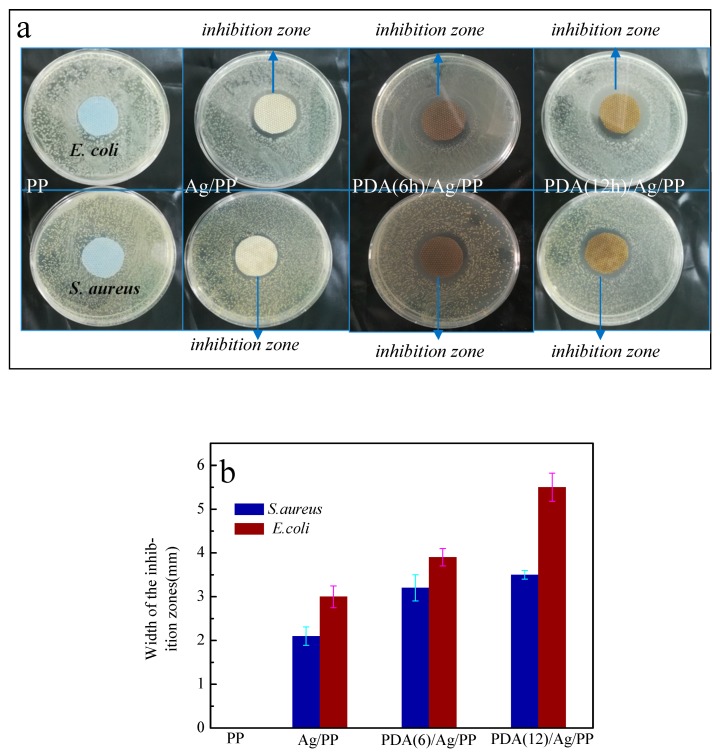
The antibacterial activity of the treated and untreated fabric: (**a**) images of antibacterial activity of the fabrics; (**b**) widths of the inhibition zone.

**Figure 10 polymers-11-00627-f010:**
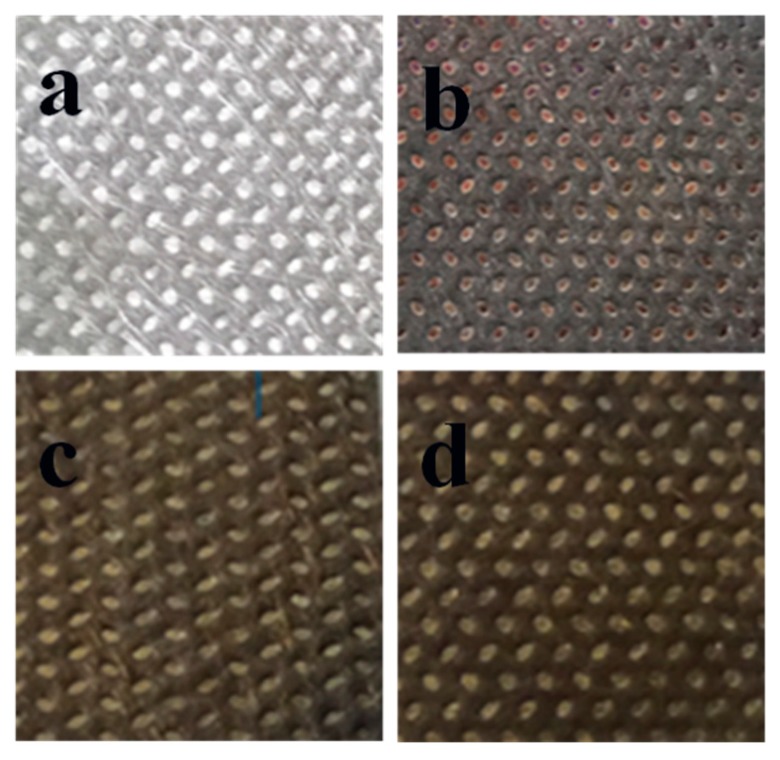
The color change of the fabric before and after the treatment of Na_2_S: (**a**,**b**) Ag-coated PP nonwoven before and after Na_2_S treatment; (**c**,**d**) PDA deposited (12 h) on the Ag-coated PP nonwoven before and after Na_2_S treatment.

**Figure 11 polymers-11-00627-f011:**
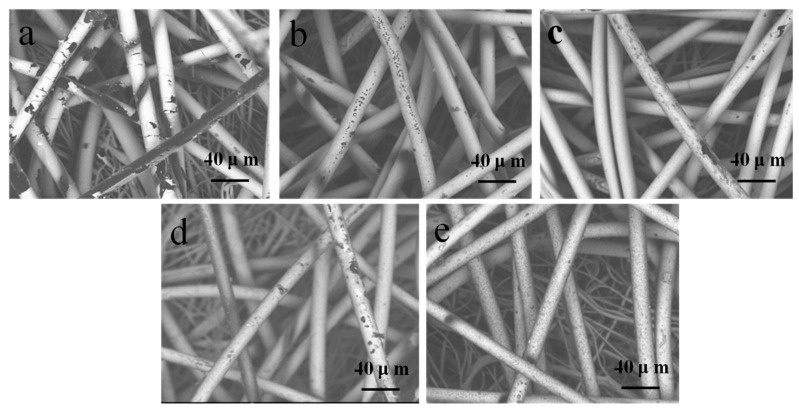
Surface morphology of samples after the treatment of Na_2_S: (**a**) Ag-coated PP nonwoven; (**b**) PDA deposited (3 h) on the Ag-coated PP nonwoven; (**c**) PDA (6 h) deposited on the Ag-coated PP nonwoven; (**d**) PDA deposited (9 h) on the Ag-coated PP nonwoven; and (**e**) PDA deposited (12 h) on the Ag-coated PP nonwoven.

**Figure 12 polymers-11-00627-f012:**
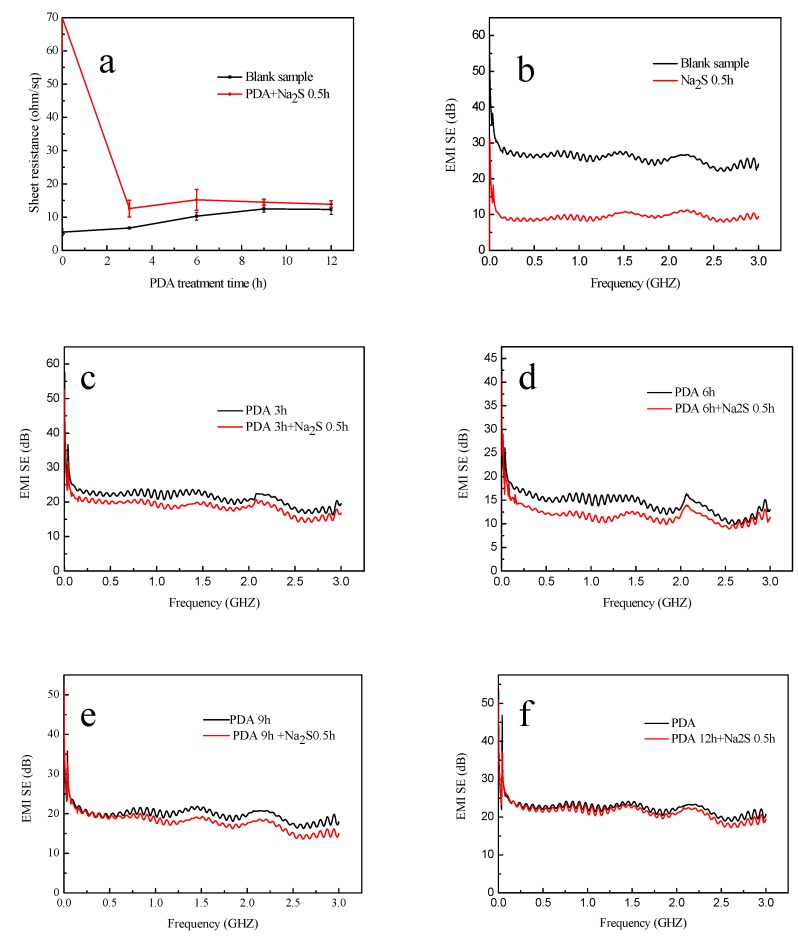
Electrical conductivity and EMI SE of the samples: (**a**) the sheet resistance of the treated samples before and after the treatment of Na_2_S; (**b**–**f**) EMI SE of the samples before and after the treatment of Na_2_S with different infiltration times of dopamine.

**Figure 13 polymers-11-00627-f013:**
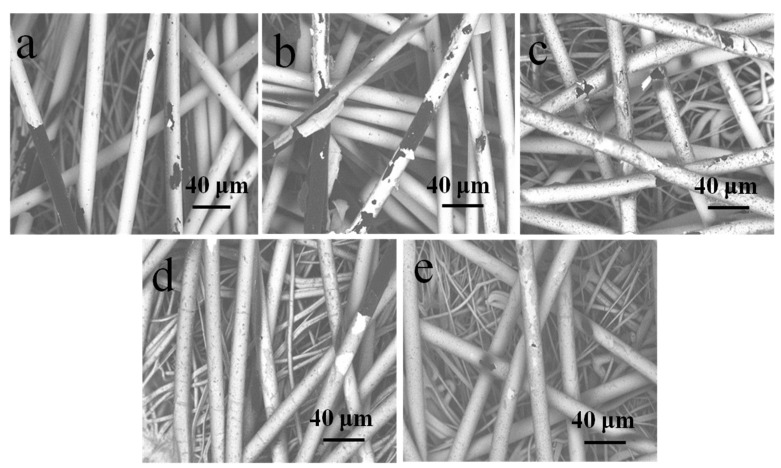
Surface morphology of samples after washing: (**a**) Ag-coated PP nonwoven; (**b**) PDA deposited (3 h) on the Ag-coated PP nonwoven; (**c**) PDA deposited (3 h) on the Ag-coated PP nonwoven; (**d**) PDA deposited (3 h) on the Ag-coated PP nonwoven; and (**e**) PDA deposited (3 h) on the Ag-coated PP nonwoven.

**Figure 14 polymers-11-00627-f014:**
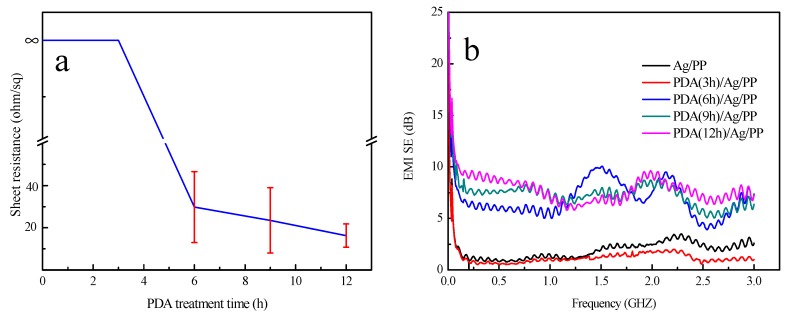
Electrical conductivity and EMI SE of the samples after washing: (**a**) sheet resistance of the treated samples and (**b**) EMI SE of the samples.

**Table 1 polymers-11-00627-t001:** Deposition parameters of Ag film.

Film	Base Pressure	DC Power	Ar Flow Rate	Working Pressure	Thickness
Ag	5.0 × 10^−4^ Pa	100 W	20 sccm	3.0 × 10^−1^ Pa	100 nm

**Table 2 polymers-11-00627-t002:** Electrical resistivity of the dopamine-treated Ag-coated samples.

PDA Treatment Time (h)	0	3	6	9	12
R (Ω/sq)	5.5	6.75	10.3	12.5	12.6

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
