# Peer review of "The Effect of Polydopamine on an Ag-Coated Polypropylene Nonwoven Fabric"

_polymers, 2019, doi:10.3390/polym11040627_

Round 1
Reviewer 1 Report
This is an interesting work on a method for preparing functional fibrous materials. One thing that it is still not so clear to me after reading this paper is how the polydopamine improves the stability of the coating. Silver is still coated on top of the PP fibers so the adhesion should be the same weak as it is in other studies. How a topcoat of PDA can improve the adhesion? Could the authors explain the mechanism of improved durability a bit better? Other than that I have a few more comments:
Introduction: “tons of researchers” is informal style of writing. It would be better to write “attracted significant attention from the scientific community” or something along these lines.
Introduction: When the authors mention the obstacles of magnetron sputtering would be nice to support them with some references.
Paragraph 3.1, line 3: should be “observe” and not “observed”
Paragraph 3.1, lines 5-6: Wrong grammar. Could be mentioned instead “throughout the fiber’s surface”
Paragraph 3.1: The elemental analysis shows a significant presence of carbon. According to the authors this is uncoated fiber or it is related to deposition of volatile organic compounds? Please include a comment in the text.
Figure 7 caption: It should be “a LED” and not “an LED”
Please increase the fonts of Figure 12. They are not visible.
General question: What is the type of bonding between dopamine and silver? Is it physical deposition or some chemical bonding is involved?
Author Response
Dear Reviewer,
Thank you very much for your comments from the reviewers about our paper submitted to polymers - 460939. Those comments are all valuable and very helpful for revising and improving our paper, as well as the important guiding significance to our researches.
Response to Reviewer #1:
Comment 1: One thing that it is still not so clear to me after reading this paper is how
the polydopamine improves the stability of the coating. Silver is still coated on top of
the PP fibers so the adhesion should be the same weak as it is in other studies. How a
topcoat of PDA can improve the adhesion? Could the authors explain the mechanism of improved durability a bit better?
Response: Thanks for your question, in this study, we immersed the Ag-coated fabeics into the dopamine solution, due to the special structure of fabric, the dopamine could deposit everywhere of the fabric such as the Ag-coating surface, the PP fibers and beetween the fibers and the Ag coating, not only on the top of Ag.
As for the improvement of adhesion, on one hand, the deposited PDA or dopamine can form noncovalent binding interaction such as metal coordination or chelating, hydrogen bonding, π−π stacking, and quinhydrone charge-transfer complexes to yield an effective adlayer with Ag. Therefore, coordination bonding and chelating bonding interactions play central roles in the adhesion of polydopamine on Ag coating. On the other hand, when in the alkaline environment, the PDA adhered to the PP fiber by covalent coupling. Because of the existence of PDA, Ag coating adhered to PP nonwoven fabrics tightly, and then the adhesion was improved.
Besides, the coating of PDA can protect the Ag layer from being in touch with outside directely, which can improve the durability of the Ag film.
And we have re-written this part according to the Reviewer ’s suggestion in the manucript, please refer to Page 22 “…that because the PDA can form covalent bond with the fabric and at the same time, it can also form coordination bonding and chelating bonding with the Ag coating, which made the Ag coating adhered to PP nonwoven fabric tightly. And the excellent adhesion plays vital role in improving the durability of the treated fabric. Also, the external PDA coating can protect the Ag film on the fabric from serious damage.” And Page 20 “…Ag coating formed metal coordination with the dopamine and with the increacing of the treatment time, the surface of the Ag coating was covered with PDA coating, which can weake the corrosion of Ag film. Therefore, the samples become much more stable because of the treatment of dopamine.”.
Comment 2: Introduction: “tons of researchers” is informal style of writing. It would
be better to write “attracted significant attention from the scientific community” or
something along these lines.
Response: We are very sorry for our incorrect writing, and we have corrected the expression “tons of researchers” as “attracted significant attention from the scientific community”. Please refer to Page 2.
Comment 3: Introduction: When the authors mention the obstacles of magnetron sputtering would be nice to support them with some references.
Response: Good suggestion, and we have added the new references in the introduction.
Jiang, S.; Miao, D.; Zhao, D. Adhesive properties of s.s to pu and pvc leathers. INT J CLOTH SCI TE. 2014, 2, 108-117.
Jiang, S.; Miao, D.; Li, A.; Guo, R. Adhesion and durability of Cu film on polyester fabric prepared by finishing treatment with polyester-polyurethane and aqueous acrylate. Fibers & Polymers. 2016, 9, 1397-1402.
Comment 4: Paragraph 3.1, line 3: should be “observe” and not “observed”.
Response: We are very sorry for our incorrect writing, “observed” was corrected as “observe”. Please refer to Page 9.
Comment 5. Paragraph 3.1, lines 5-6: Wrong grammar. Could be mentioned instead
“throughout the fiber’s surface”
Response: We are very sorry for our negligence of the wrong grammar, we have added “ fiber’s ” into the sentence, please refer to Page 9.
Comment 6: Paragraph 3.1: The elemental analysis shows a significant presence of carbon. According to the authors this is uncoated fiber or it is related to deposition of volatile organic compounds? Please include a comment in the text.
Response: Thanks for your question, Fig 3b demonstrated the elemental mapping image of the sputtered fabric, C is the main element of polypropylene fiber,so the C element was obtained from polypropylene fiber, and the uncoated polypropylene fiber contributed to the proportion 37.58% of C. And because the fiber was coated with Ag film, the proportion of C is much less than Ag. The existence of O may result from the Ag oxidation. Therefore, as Reviewer suggested that we have included a comment in the text, please refer to Page9:
“Furthmore, the proportion of C is 36%, which was obtained from the polypropylene fibers. And the oxidation of Ag resulted in the proportion 3% of O.”
Comment 7: Figure 7 caption: It should be “a LED” and not “an LED”.
Response: Sorry for our negligence of the grammar mistake, and we have revised “an LED” into “a LED”, we can find it in Page14.
Comment 8: Please increase the fonts of Figure 12. They are not visible.
Response: We have made correction according to the Reviewer ’s comments. We have increased the fonts of Figure 12.
、
Comment 9: General question: What is the type of bonding between dopamine and silver? Is it physical deposition or some chemical bonding is involved?
Response: Thanks for your question, after studied several literatures, we have known that on inorganic surfaces such as metal, the unoxidized dopamine forms high-strength yet reversible coordination bonds. And the details, you can refer the response to the comment 1. So, the type of bonding between dopamine and silver is chemical bonding.
Reviewer 2 Report
The manuscript could be accepted after revised from following comments.
1) In the introduction part, the authors should add the following suggested references, regarding conductive fibres and yarns. "Polymer Interface Molecular Engineering for E-Textiles." Polymers 10.6 (2018): 573. And "Mussel‐Inspired Flexible, Durable, and Conductive Fibers Manufacturing for Finger‐Monitoring Sensors." Advanced Materials Interfaces (2018): 1801547.
2) Dopamine which mimics the adhesive chemistry of mussel plaque detachment allows the spontaneous deposition of nanoscale-thin, surface-adherent films of poly(dopamine) (PDA) on virtually all material surfaces by simple dip-coating in an alkaline solution. The alkaline environment should adversely affect the Ag coating on the PP fibres, resulting in a decrease of conductivity. The morphology change of Ag coating after PDA coating should be examined.
3): For analysing electromagnetic interference shielding effectiveness with varying dopamine treatment time, the treatment time of 5 min should be needed because of the rapid reaction of dopamine in the alkaline solution.
Author Response
Dear Reviewer
Thank you very much for comments from the reviewers about our paper submitted to polymers - 460939. Those comments are all valuable and very helpful for revising and improving our paper, as well as the important guiding significance to our researches.
Response to Reviewer #2:
Comment 1: In the introduction part, the authors should add the following suggested references, regarding conductive fibres and yarns. "Polymer Interface Molecular Engineering for E-Textiles." Polymers 10.6 (2018): 573. And "Mussel‐Inspired Flexible, Durable, and Conductive Fibers Manufacturing for Finger‐Monitoring Sensors." Advanced Materials Interfaces (2018): 1801547.
Response: Thanks for your suggestion, we have added the references in the manuscript, which make the introduction more complete.
Zhu, C.; Li, Y.; Liu, XQ. Polymer interface molecular engineering for E-Textiles. Polymers. 2018, 6, 573-591.
Zhu, C.; Guan XY.; Wang X.; Li, Y.; Chalmers, E. Mussel-Inspired Flexible, Durable, and Conductive Fibers Manufacturing for Finger-Monitoring Sensors. Adv. Mater. Interfaces. 2019, 1, 1801547-1801556.
Comment 2: Dopamine which mimics the adhesive chemistry of mussel plaque detachment allows the spontaneous deposition of nanoscale-thin, surface-adherent films of poly(dopamine) (PDA) on virtually all material surfaces by simple dip-coating in an alkaline solution. The alkaline environment should adversely affect the Ag coating on the PP fibres, resulting in a decrease of conductivity. The morphology change of Ag coating after PDA coating should be examined.
Response: Thanks for your question, as for the decrease of conductivity, we think it is the PDA coating that on the surface of Ag hinder the electronic transmission. After all, Ag has stable physical and chemical properties, when treated with dopamine, the alkaline environment is about 8.5, which has little influence on the conductivity.
Please refer to Page 14:
“It is obvious that the sheet resistance increased with the increasing treatment time of dopamine because of the obstruction of electronic transmission of the PDA coating.”
After the coating of PDA, in the macro level, the color of the Ag coating has changed with the treatment time of dopamine which was shown in Fig 10(a)(c). In microscopic level, the Ag coating was covered with PDA particles, we can see from Fig 6 in the manuscript.
Comment 3: For analysing electromagnetic interference shielding effectiveness with varying dopamine treatment time, the treatment time of 5 min should be needed because of the rapid reaction of dopamine in the alkaline solution.
Response: As Reviewer suggested that we have conducted the experiment which the treatment time of 5 min, and its electromagnetic interference shielding effectiveness is ~26 dB which is almost the same as the single Ag-coated fabric. However, the fabric treated for 5 minutes has poor EMI SE after Na2S and washing treatment, which are 10 and 2 dB, respectively. 5 min treatment time of dopamine make little differernce on improving the stability and durablity compared with single Ag coated fabric, so we haven’t put the treatment time into the manuscript.
Please find more detailed responses (some figures included) in the attached PDF. Thank you.

Reviewer 3 Report
Liu et al. study the influence of the deposition time of polydopamine on silver-coated polypropylene fibres. A recent work reported a similar procedure and application (Zhou, Y., Jiang, L., Guo, Y., Sun, Z., Jiang, Z., Chen, S., … Jerrams, S. (2019). Rapid fabrication of silver nanoparticle/polydopamine functionalized polyester fibers. Textile Research Journal. https://doi.org/10.1177/0040517519826893). I think the manuscript has some interesting data, but there is a lack of significant experimental justification for the claims made. As a result, I do not think the manuscript is publishable in its current form.
- The title suggests that there is a strong influence of the PDA treatment time on the behaviour of Ag-coated samples, and I believe that is not supported by the results. The antibacterial activity was only study at one deposition time; there are not significant differences in the electromagnetic shielding effectiveness; slight differences are shown in the stability tests, electrical conductivity or laundering durability.
- The manuscript lacks in quantification or scientific evidence in its studies. Some examples are: i) the authors claim that “more and more PDA particles” are deposited and they become “larger and larger. This is not supported by scientific evidence; ii) the inhibition zones for the antibacterial areas should be quantified; iii) the authors claim that the fibres display “perfect conductivity” after silver coating; etc.
- The introduction is weak in highlighting the importance of the research and the literature review is rather poor. Some claims are not supported with references and some others I cannot agree with. For instance, what do the authors mean when they attempt to relate the local surface plasmon resonance of silver particles with their antibacterial activity? Also, the justification to do this study using polypropylene fibres is quite ambiguous. Does this study apply to other types of fibres? If yes, why? If not, why not? Does the surface chemistry make an influence?
- If the aim of the study is to analyse the differences in the PDA layer, the authors should attempt to quantify the differences in thickness or particle size, or how the deposition time may affect this polydopamine layer. Also, the authors claim PDA deposit forming particles but there is no evidence in the manuscript, nor references provided.
- The authors do not mention or acknowledge the expected differences in the properties of the material on one side or another. Sputtering magnetron is a very directional deposition process, meaning that the thickness of the metal deposited won’t be uniform around the fibres. Actually, one would expect that only the most external/superficial surfaces will be coated. That means, the thickness measured (100 nm) is most probably not the thickness of the Ag coating the samples. This would affect the conductivity measurements on one or another side and also the electromagnetic interference shielding effect.
- The FITR analysis is rather poor and vague. The authors did not attempt to identify or assign characteristic vibration frequencies from the polypropylene fibres. Also, after coating with PDA, the assigned vibrations are not supported by any reference. More importantly, the 1520 cm-1 peak that the authors daringly assign to two different bonds (i.e. C=H and N-H) cannot be seen from the spectrum in the figure. If anything, it looks more like a minimum rather than a maximum in transmittance.
- The figures and particularly their captions should be improved. Captions are incomplete (e.g. inset not described in Fig. 5) and vague (e.g. Fig. 6). Fig. 3b does not show clear colours that can be related to the element composition. The inhibition zone in Fig. 9 is not clear and should be highlighted.
Author Response
Dear editor and reviewer ,
Thank you very much for your letter and the comments from the reviewers about our paper submitted to polymers - 460939. Those comments are all valuable and very helpful for revising and improving our paper, as well as the important guiding significance to our researches.
We have checked the manuscript and revised it according to the comments. We submit here the revised manuscript as well as a list of changes.
If you have any question about this paper, please don’t hesitate to let me know.
Comment 1: Liu et al. study the influence of the deposition time of polydopamine on silver-coated polypropylene fibres. A recent work reported a similar procedure and application (Zhou, Y., Jiang, L., Guo, Y., Sun, Z., Jiang, Z., Chen, S., … Jerrams, S. (2019). Rapid fabrication of silver nanoparticle/polydopamine functionalized polyester fibers. Textile Research Journal.https://doi.org/10.1177/0040517519826893). I think the manuscript has some interesting data, but there is a lack of significant experimental justification for the claims made. As a result, I do not think the manuscript is publishable in its current form.
Response: We have added the “Zhou, YF.; Jiang, L.; Guo, Y.; Sun, ZH. Rapid fabrication of silver nanoparticle /polydopamine functionalized polyester fibers. TEXT RES J. 2019. 1-11.” in the manuscript as a reference.
The recent work reported that Zhou et al. modified PET fibers with dopamine and followed by electroless plating to fabricate silver coated PET fibers. They focused on the reducing properties of dopamine which can reduce the Ag+ into Ag nanoparticles, and deposited on the surface of PET fibers.
In our manuscript, we used magnetron sputtering system to form an Ag film on the surface of the fabric, and then the coated fabric was treated with dopamine. We take advantage of the strong adhesion not the reducing property of dopamine.
Also, we are sorry for the lack of significant experimental justification for the claims made. Considering your comment, we have re-written the claims made in the manuscipt for example:
In page 11: “As illustrated in the SEM images, the surface of the fibers prone to be rougher that is because of the deposition of the PDA particles. Moreover, the particle size is from 0 um up to 0.490 um with the treatment time increased, which was shown in Fig 5a. And the PDA coating formed finally.” The statement was added.
In page 20: The explanation has been re-written “Ag coating formed metal coordination with the dopamine [35] and with the increacing of the treatment time, the surface of the Ag coating was covered with PDA coating, which can weaken the corrosion of Ag film. Therefore, the samples become much more stable because of the treatment of dopamine.”
In page 22: We have added the statement “that because the PDA can form covalent bond with the fabric [14] and at the same time, it can also form coordination bonding and chelating bonding with the Ag coating, which made the Ag coating adhered to PP nonwoven fabric tightly. And the excellent adhesion plays vital role in improving the durability of the treated fabric. Also, the external PDA coating can protect the Ag film on the fabric from serious damage.”
In page 19: “This image indicates that electrical conductivity of the single Ag-coated sample declined rapidly and the sheet resistance was about 70 Ω/sq because of the broken Ag film.”
And the other revisions have been highlighted in the manuscript.
Comment 2: The title suggests that there is a strong influence of the PDA treatment time on the behaviour of Ag-coated samples, and I believe that is not supported by the results. The antibacterial activity was only study at one deposition time; there are not significant differences in the electromagnetic shielding effectiveness; slight differences are shown in the stability tests, electrical conductivity or laundering durability.
Response: Thanks for your question, we changed the title into “The influence of dopamine on the Ag coated polypropylene nonwoven fabrics”
In the manuscript, we conducted the experiment that treated the Ag coated fabric with dopamine in different time. The results showed that after the treatment of dopamine, the fabric has changed little in conductivity and electromagnetic shielding effectiveness. After that, we treated the fabric with Na2S and laundering, and found that the electrical conductivity and the electromagnetic shielding effectiveness of single Ag-coated fabric dropped seriousely, but the fabric treated with dopamine didn’t change significantly. Therefore, we conclude that the existence of PDA has a positive influence on protecting the Ag coating.
As the Reviewer mentioned, we have studied at different deposition time, and added the date in Page 17. And we have rewrited the date analysis. Please refer Page 16 :
“Moreover, the widths of inhibition zone (shown in Fig 9b) of the dopamine - treated (6 and 12 h) Ag-coated fabrics were measured to be 3.2 and 3.5 mm for S. aureus, and 3.9 and 5.5 mm for E.coli, respectively, which were larger than that of the single Ag-coated fabrics (2.1 and 3.0 mm in width for S. aureus and E.coli, respectively). Thus, it reveals that the treatment of dopamine can enhance the antibacterial activity of Ag-coated fabrics.”
Comment 3: The manuscript lacks in quantification or scientific evidence in its studies. Some examples are: i) the authors claim that “more and more PDA particles” are deposited and they become “larger and larger. This is not supported by scientific evidence; ii) the inhibition zones for the antibacterial areas should be quantified; iii) the authors claim that the fibres display “perfect conductivity” after silver coating; etc.
Response: Thanks for your question, we are aware of the shortcomings in the manuscript and add some experimental data to make a more scientific explanation.
i) We used the weight gain and the particle size to quantify PDA coating. The particle size is from 0 um up to 0.490 um with the treatment time increased, which was shown in Fig 5a. And the PDA coating formed finally. The weight gain increased with the increasing treatment time of dopamine, the weight gain was up to 0.65 g/m2 when the treatment time was 12 h.
ii) According to the Reviewer’s comment, the inhibition zones for the antibacterial areas has been quantified. And they are quantified by the widths of inhibition zone. Please refer Page 16 :“Moreover, the widths of inhibition zone (shown in Fig 9b) of the dopamine - treated (6 and 12 h) Ag-coated fabrics were measured to be 3.2 and 3.5 mm for S. aureus, and 3.9 and 5.5 mm for E.coli, respectively which were larger than that of the single Ag-coated fabrics (2.1 and 3.0 mm in width for S. aureus and E.coli, respectively). Thus, it reveals that the treatment of dopamine can enhance the antibacterial activity of Ag-coated fabrics.”
iii) According to the Reviewer’ comment, the description “perfect conductivity” was replaced by “Ag possesses electrical conductive of properties of approximately 1.59 × 10 −8 Ω m at 293 K.” in Page 14.
Moreover, we have revised the shortcoming, , and highlighted it in the manuscript.
Comment 4: The introduction is weak in highlighting the importance of the research and the literature review is rather poor. Some claims are not supported with references and some others I can not agree with. For instance, what do the authors mean when they attempt to relate the local surface plasmon resonance of silver particles with their antibacterial activity? Also, the justification to do this study using polypropylene fibres is quite ambiguous. Does this study apply to other types of fibres? If yes, why? If not, why not? Does the surface chemistry make an influence?
Response: i) As for the antibacterial activity, there is something wrong with the description, so we deleted some of them, which stated as follows: such as very low resistivity among all metals and antibacterial activity.
ii) we choosed polypropylene nonwoven fabric as subtract because (1). The nonwoven fabric can own electrical conductivity after Ag was sputtered onto it for its continuous surface, which was shown in Fig 3a; (2). Polypropylene fibres have stable chemical property, which can withstand the enviroment of pH = 8.5 when treated with dopamine. We have revised in the manuscript in Page 4 “…its special continuous surface structure that can be conductive after sputtered, and stable chemical properties especially the alkali resistance.”, and in Page 9 “As we can see, the polypropylene fibers were intertwined irregularly, so that there were enough nodes between fibers, which could ensure the electrical conductivity of the fabric after magnetron sputtering process.”.
And this study can apply to other types of fibers which process the resistance of alkali. However, only when the surface of the fabric has continous surface, the fabric can be conductive after sputtered.
When treated with dopamine, the condition was fixed at pH = 8.5. Only the fibers have the alkali resistance, can it exempt from broken. In conclusion, the surface chemistry make an influence.
Comment 5: If the aim of the study is to analyse the differences in the PDA layer, the authors should attempt to quantify the differences in thickness or particle size, or how the deposition time may affect this polydopamine layer. Also, the authors claim PDA deposit forming particles but there is no evidence in the manuscript, nor references provided.
Response: The aim of the study is to analyse the influence of different PDA coating on Ag sputtered PP nonwovens, hence we quantify the differences in particle size in Fig 5a. With the treatment time increased, the PDA particles size became larger, and the weight gain increaced which demonstrated more particles deposited onto the fabric, and the PDA coating was formed finally.
As for the PDA particles, we have observed the surface morphology of the dopamine treated fabric which was shown in Fig 6.
Comment 6: The authors do not mention or acknowledge the expected differences in the properties of the material on one side or another. Sputtering magnetron is a very directional deposition process, meaning that the thickness of the metal deposited won’t be uniform around the fibres. Actually, one would expect that only the most external/superficial surfaces will be coated. That means, the thickness measured (100 nm) is most probably not the thickness of the Ag coating the samples. This would affect the conductivity measurements on one or another side and also the electromagnetic interference shielding effect.
Response: Thanks for your question, we haven’t described the sputtered process clearly in the manuscript. In this study we only sputtered one side of the most external surfaces of the fabric, so only one side of the fibers were coated with Ag, we changed in the manuscript “The Ag film was deposited on one side of the PP nonwoven fabrics at room temperature by magnetron sputtering apparatus.” in Page 5.
And the thickness of Ag coating is 100 nm which was measured by online measurement system and the thickness is used to characterized the sputtered surface. And the sputtered side of the fabric was carried out conductivity measurements and electromagnetic interference shielding effect.
Comment 7: The FITR analysis is rather poor and vague. The authors did not attempt to identify or assign characteristic vibration frequencies from the polypropylene fibres. Also, after coating with PDA, the assigned vibrations are not supported by any reference. More importantly, the 1520 cm-1 peak that the authors daringly assign to two different bonds (i.e. C=H and N-H) cannot be seen from the spectrum in the figure. If anything, it looks more like a minimum rather than a maximum in transmittance.
Response: We have done the FITR several times and haven’t found the ideal spectrum, so we decided to apply the analysis of EDS to prove the existance of PDA.
EDS mapping dots analysis and SEM image of the dopamine treated Ag-coated fabric were displayed in the Fig 6. The mapping dots of C, O, N and Ag elements prone to have a homogeneous distribution. Compared with Fig 3b, the kind of the elements as well as the proportion changed. The PDA possesses C, O and N elements, so the element of N was totally abtained from PDA and it was distributed homogeneously on the surface of the fiber which proved the PDA evenly distributed in the fiber. Moreover, the amount of O was up to 18% due to the O element in PDA and the Ag oxidation. Therefore, it can be easily concluded that the fiber was deposited with PDA. As shown in the SEM images, the PDA paritcles deposited onto the Ag coating, and particle size is about 0.49 um in the treatment time of 12 h.
Fig 6. The mapping dots and elements weights of the dopamine treated Ag-coated fabric.
Comment 8: The figures and particularly their captions should be improved. Captions are incomplete (e.g. inset not described in Fig. 5) and vague (e.g. Fig. 6). Fig. 3b does not show clear colours that can be related to the element composition. The inhibition zone in Fig. 9 is not clear and should be highlighted.
Response: Thanks for your conment, we have made some changes according to the comment:
i) We have completed the captions (e.g. “(b) the color change of the dopamine solution) and others you can find in the manuscript.
ii) We have replaced the Fig 6 with a new one.
iii) The inhibition zone in Fig. 9 has been highlighted.
Please find more detailed responses (some figures included) in the attached PDF. Thank you.

Round 2
Reviewer 2 Report
Suggested title: The effect of polydopamine on the Ag-coated polypropylene nonwoven.
The word effect is used in experimental research, while the word influence is used in survey type research.
Author Response
Dear editor and reviewer,
Thank you very much for your letter and the comments from the reviewers about our paper submitted to polymers - 460939. Those comments are all valuable and very helpful for revising and improving our paper, as well as the important guiding significance to our researches.
We have checked the manuscript and revised it according to the comments. We submit here the revised manuscript as well as a list of changes.
If you have any question about this paper, please don’t hesitate to let me know.
Sincerely yours,
Dr. Miao
Response to Reviewer #2:
Comment: Suggested title: The effect of polydopamine on the Ag-coated polypropylene nonwoven.
The word effect is used in experimental research, while the word influence is used in survey type research.
Response: Good suggestion, we have revised the title as the Reviewer suggested, and changed in the manuscript.
Reviewer 3 Report
The authors have addressed the comments and have significantly improved the first version of the manuscript. I believe it is now publishable in its present form.
Author Response
Dear editor and reviewer,
Thank you very much for your letter and the comments from the reviewers about our paper submitted to polymers - 460939. Those comments are all valuable and very helpful for revising and improving our paper, as well as the important guiding significance to our researches.
We have checked the manuscript and revised it according to the comments. We submit here the revised manuscript as well as a list of changes.
If you have any question about this paper, please don’t hesitate to let me know.
Sincerely yours,
Dr. Miao
This manuscript is a resubmission of an earlier submission. The following is a list of the peer review reports and author responses from that submission.
Round 1
Reviewer 1 Report
The paper deals with Ag sputtering on PP nonwoven followed by dip coating in dopamine. The results are, however, presented rather in the style of a technical report of data than a scientific paper.
Apart from the missing sound scientific discussion, there are few points making the paper not publishable in polymers
- Wrong chemical formula in Fig. 2
- Polymerisation mechanism of dopamine cannot be found in Ref. 32
- Method to determine weight gain (Fig. 5) is missing
- The identification of PDA using IR as reported is very vague (exact description of vibration bands, reference of PDA and PP/PDA samples)
- How is the contacting in Fig. 7 with LED bulb (on the surface of through the substrat)
- Metal coordination mechanism Ag / PDA cannot be found in Ref. 35, 36
- Covalent bond between PDA with PP fabric (and Ag film) cannot be found in Ref. 17
Furthermore, the manuscript is focusing rather on deposition and coating than on polymers. Considering another (more suitable) journal is recommended after a complete revision.
Author Response
Thank you for your professional comments on our paper. We have revised our paper according to your comments, please refer to the appendix.

Reviewer 2 Report
This work by Liu et al. describes a surface modification method PP fabrics based on a combination of dopamine and Ag. The results do not any significant novelty. The properties of Ag as an antibacterial medium are known, as well as those of dopamine. The results also do not show any exceptional performance compared to the state-of-the-art. However, I should admit that the authors have done a detailed characterization of the material which can be a useful reference for relevant future studies. Finally the level of English should be significantly improved and in the abstract and conclusions some more quantitative results should be mentioned rather than qualitative statements. I would suggest that this manuscript should be significantly revised before being considered for publication. A few more comments follow:
1) Line 17: It says “deposited” but from the context I believe that “coated” should be the correct word here.
2) Line 22: In the abstract the improvement of laundering durability should be quantified.
3) Line 41: What are the aesthetic values exactly? Not clear to me.
4) The schematic in Figure 1 could be improved. What is the difference between the 2nd and 3rd step for example in terms of graphics? How is the film formation depicted?
5) Line 152: It should be better “the surface of the fibers” and not “the surfaces of the fiber”.
6) Line 175: There are some syntax errors here.
7) In the conclusions it would be good to also provide some numbers and not only qualitative comments.
Author Response

(The authors gave the same response as above.)

Round 2
Reviewer 1 Report
The authors have revised the manuscript in order to cope with reviewer’s comments.
However, the answers / comments are not sufficient:
- New referent 32 only shows a proposed structure of polydopamin, but not the “formation process of PDA” as given by the authors. Actually, the mechanism of polymerization of dopamine is still a subject which is still not fully understood
- The authors argued that the IR spectrum of PP/PDA was the same as the untreated PP nonwoven fabric, maybe due to the too small, non-detectable PDA content: a) how is the spectrum looks like ? and b) why can the deposition of DPA on Ag-coated PP be detected by IR ?
- Regarding covalent bond between PDA with PP fabric and Ag film: PP in unmodified form does not bear any reactive groups for any reaction with PDA. Ag as noble metal cannot form any covalent bond with dopamine. The authors should study Ref. 14 and/or others thoroughly before making such statements